# Effect Application of Apple Pomace on Yield of Spring Wheat in Potting Experiment

Marcin Różewicz *, Marta Wyzińska and Jerzy Grabiński

The Institute of Soil Science and Plant Cultivation—State Research Institute, ul. Czartoryskich 8, 24-100 Pulawy, Poland; mwyzinska@iung.pulawy.pl (M.W.); jurek@iung.pulawy.pl (J.G.)
* Correspondence: mrozewicz@iung.pulawy.pl

**Abstract:** Apple pomace, as a by-product, is difficult to manage and is produced in significant quantities. This makes it necessary to manage the resulting biomass. It is important for the environment to use pomace in an ecological way. It can provide a source of organic matter and be composted, but it can also be added directly to the soil. The greatest impediments in the use of pomace are the constant process of its production and the fermentation processes taking place within it, which require storage of action and drying and transportation of the pomace. Using pomace immediately after its formation as an exogenous source of organic matter for soil is a possibility. This method of pomace management benefits society and the natural environment. Thus, a study was undertaken to determine the feasibility of applying apple pomace to soil in a model experiment. Tests were conducted on spring wheat of the Harenda cultivar in a greenhouse. Various amounts of apple pomace were added to the soil. Soil properties were studied, as well as photosynthetic parameters and crop yield structure. It was shown that it is possible to improve soil properties and plant yield by adding pomace to the soil, but only for a limited quantity of pomace. The highest maximum pomace that should be used, for spring wheat in field conditions, is a maximum of 2 t/ha$^{-1}$. At this dose of apple pomace, the characteristics of the wheat yielding structure were significantly improved, such as plant tillering, the number of ears and the weight of kernels per spike, and the weight of a thousand kernels. Since this was a model experiment, it should be treated as an introduction to research on the use of pomace, and further research on the possibility of using pomace in field conditions, including for other cereal species, should be continued.

**Keywords:** apple pomace; soil quality; photosynthesis efficiency; waste fertilizers; organic matter

## 1. Introduction

Poland is one of the largest apple producers in Europe. A significant proportion of the fruit is used for juice production. The pressing of raw fruit produces juice and a by-product in the form of pomace. The increasing amount of waste generated in fruit processing, which includes seeds, pulp residues, and peel, creates a global environmental and economic problem. The considerable scale of juice and concentrate production results in a large amount of apple pomace, which should be used in some way. Some of it is reused in the food industry [1–5]. It can also be partially used as animal feed, but only in a limited amount, as exceeding the allowed proportion in the feed results in poorer animal performance [6,7]. The large scale of production, its limited use in the food and feed industry, and the significant water content of the pomace make it difficult to transport and result in a rapid spoilage process, necessitating the search for economic and logistical solutions favouring its secondary management. It is therefore necessary to carry out research that will contribute to a better use of pomace for the benefit of the environment, entrepreneurs, and also consumers. The biggest impediment to the use of pomace is the continuous production process and the fermentation processes that take place in the pomace, which necessitate storage of the raw material and its drying and transport. One

possible solution is to manage pomace as an exogenous source of organic matter for the soil. This type of pomace management would result in benefits for society and the environment. Apple pomace contains significant amounts of pectin, which binds water and can effectively retain it in the soil. Pomace also has the advantage of containing valuable elements that would return to the soil, as well as phytochemicals that can have a positive effect on the soil and plants. The phytochemical components of apple pomace have antifungal effects against some plant pathogens [8,9]. In addition, pectin contained in apple pomace can protect plants from absorbing harmful heavy metals and accumulating them in their tissues [10]. The use of pomace additions to soil can have multifaceted positive effects on both plants and soil. Currently, there is little research on the potential for direct addition of apple pomace to soil.

A study was therefore undertaken to determine what effect the application of different amounts of apple pomace to the soil has on wheat plants and their yield. The aim of the study was to test the feasibility of applying pomace to the soil and to determine the optimum dose of apple pomace that would have the most beneficial effect on the plants. It was assumed that there is an optimal level of pomace addition, which positively affects soil properties and the plants themselves through improving their physiological parameters and yield.

## 2. Materials and Methods

### 2.1. Plant Growth Conditions

A three-year pot experiment was conducted from 2019–2021 at the greenhouse of the Institute of Soil Science and Plant Cultivation—State Research Institute in Puławy [51°24′59″ N 21°58′09″ E]. Spring wheat of the Harenda cultivar was used in the experiment. This cultivar is characterised by high grain yield and quality, and resistance to diseases and lodging. The pot experiment was set up using Mitscherlich pots, which contained 7 kg of soil. The experimental factor was the level of apple pomace applied. Four factor levels were used: K—control with no pomace added to the soil, D1—pomace addition of 70 g/pot (corresponding to a field application rate of 1 t·ha$^{-1}$), D2—addition of 140 g/pot (corresponding to a field application rate of 2 t·ha$^{-1}$), and D3—addition of 210 g/pot (corresponding to a field application rate of 3 t·ha$^{-1}$). The composition of the pomace is shown in Table 1.

**Table 1.** Content in apple pomace (g/kg$^{-1}$ dry matter) of nitrogen (N), carbon (C), potassium (K), and phosphorus (P).

| Specification | N | C | K | P |
|---|---|---|---|---|
| Content (g/kg$^{-1}$ of dry matter) | 10 | 80 | 1.7 | 7.0 |

Pre-sowing macro and micronutrient fertilisation was applied at a rate of:

- $P_2O_5$—2.52 g/pot as $KH_2PO_4$,
- $K_2O$—2.04 g/pot as $K_2SO_4$,
- Mg—0.5 g in the form of $MgSO_4$
- Fe 50 mg/pot,
- Br 5 mg/pot,
- Mn 3 mg/pot,
- Cu 3 mg/pot.

Nitrogen fertilisation was applied at a rate of 2.4 g N/pot in the form of $NH_4NO_3$ using $\frac{1}{2}$ dose before sowing and $\frac{1}{2}$ dose at the stalk shooting stage.

Sowing was performed at the optimum date for spring wheat, sowing 20 grains of wheat in each pot, leaving 10 of the strongest plants after emergence, which were maintained until harvest. Plant emergence in all pots was determined at the staking stage.

## 2.2. Gas Exchange Properties of the Soil

Soil gas exchange parameters were measured twice with a CIRAS-2 model instrument (PP Systems, Amesbury, MA, USA), a portable infrared $CO_2$ analyser equipped with a Soil $CO_2$ Flux dynamic with a closed chamber. The closed chamber has a top volume of 1.17 $dm^3$, covered a soil area of 75.6 $cm^2$ and was held in the soil during the measurement.

Gas exchange parameters were estimated based on the growth rate of $CO_2$ in a closed chamber over a period of 60 s. The measurement was performed on each pot.

All measurements were made during the day in the morning. Four gas exchange parameters were determined.

- Soil respiration (Sr)
- Evapotranspiration (Er)
- Changes in carbon dioxide concentration ($\Delta CO_2$)
- Changes in soil moisture ($\Delta MB$)

## 2.3. Photosynthesis Parameters

The photosynthesis measurements made were performed with a CIRAS-2 Portable Infrared Gas Exchange Analyser Type CIRAS-2 Portable Photosynthesis System (Hitchin, Herts, UK).

The parameters measured were:

- Net photosynthetic efficiency Pn ($\mu mol\ m^{-2}\ s)^{-1}$
- Transpiration efficiency ($mmol\ m^{-2}\ s)^{-1}$
- Stomatal conductivity gs ($mmol\ H_2O\ m^{-2}\ s)^{-1}$
- Intracellular carbon dioxide concentration Ci ($\mu mol\ CO_2\ m^{-2}\ s)^{-1}$

Photosynthesis was measured at the flag leaf stage in four replicates at each experimental site. A measuring cell was placed on the flag leaf with a light intensity of 1000 PAR ($\mu mol\cdot m^{-2}\cdot s^{-1}$) supplied by a light unit attached to the cuvette. The following conditions were assumed in the analyser cuvette: a constant supply of carbon dioxide equal to 370 ppm ($\mu mol\ CO_2\cdot mol^{-1}$ of air), humidity equal to the ambient humidity, air temperature equal to +25 °C. For each experimental plot, a measurement was taken at the centre of four randomly selected plants. In addition, the photosynthetic water use efficiency (WUE) was determined from the ratio of photosynthesis to transpiration intensity (Pn/E).

The SPAD leaf greening index was determined using the SPAD-502 chlorophyll meter (Minolta Co., Ltd., Ramsey, NJ, USA). The value of each measurement was the average of 30 measurements taken on a wheat flag leaf.

## 2.4. Determination of Seed Yield and Structural Characteristics

At full maturity, the plants were harvested, and yield structure traits were analysed: number of ears, grain weight per ear, grain weight per pot, thousand-grain weight, and straw weight. Harvest index was calculated on the basis of grain weight divided by the sum of grain and straw weights.

## 2.5. Statistical Analysis of the Data

The analytical data obtained were the average of 4 pots from the same site. Statistical analysis of the results obtained was performed with Statistica v.13.1. using the Tukey test at $p \leq 0.05$.

## 3. Results and Discussion

### 3.1. Gas Exchange Properties of the Soil

The analysis of the results showed that, with an increase in the addition of apple pomace to the soil, the soil respiration rate decreases significantly. This is beneficial due to the reduction of oxygen access and the slowing of organic matter decomposition and, thus, carbon dioxide emissions to the atmosphere [11]. The reason for the reduced soil respiration with the increased addition of pomace to the soil may have been due to its

action of clumping soil particles into larger aggregates. The larger soil aggregates formed by the addition of pomace create conditions that impede gas exchange [12]. Reduced soil respiration could also be caused by a disruption of the C:N ratio due to the introduction of more organic carbon into the soil. In the case of straw introduced into the soil as organic matter, it can be concluded that it has a positive effect on soil respiration properties but exceeding the optimum addition of straw results in reduced soil respiration [11]. The control site and site D1, where the lowest level of pomace addition to the soil was applied, did not differ significantly from each other and showed the lowest evapotranspiration. The other two sites, where a higher proportion of pomace was applied, showed significantly higher evapotranspiration values. As shown by Nosalewicz et al. [12], the addition of apple pomace to the soil can negatively affect plants through faster evaporation of water from the soil, thus reducing the amount of water available to plants. Our research on wheat also confirms that, when irrigation was applied equally to all sites, there was a negative effect on plants at site D3 with the highest soil addition, which, based on the results, could be related to faster evaporation of water from the soil. Changes in carbon dioxide concentration were significantly differentiated by the addition of apple pomace to the soil. The highest proportion of pomace resulted in the highest concentration of $CO_2$ in the soil. Each of the sites differed significantly, with the largest difference found between site D1 and D2, which was 57%. Between sites D2 and D3, there was a smaller difference of 20%. The results of changes in soil moisture showed no significant differences between the control treatment (K), where no apple pomace was added, and treatments D1 and D2. A significant difference was found at site D3, where the highest addition of apple pomace was applied (Table 2).

**Table 2.** Soil gas exchange parameters as a function of pomace addition to the soil.

| Specification | K | D1 | D2 | D3 |
|---|---|---|---|---|
| Soil respiration (Sr (g $CO_2$/m/h) | 0.27 [a] | 0.25 [b] | 0.20 [c] | 0.18 [d] |
| Evapotranspiration (Er g $CO_2$/m/h) | 34.59 [a] | 36.59 [a] | 50.06 [b] | 66.89 [c] |
| Changes in carbon dioxide concentration ($\Delta CO_2$) | 28.92 [d] | 35.70 [c] | 62.71 [b] | 72.6 [a] |
| Changes in soil moisture ($\Delta MB$) | 2.90 [b] | 2.80 [b] | 2.90 [b] | 3.10 [a] |

Different letters (a–d) are significantly different ($\alpha = 0.05$).

### 3.2. Photosynthesis Parameters

The net photosynthesis value was significantly higher in wheat from treatments D1 and D2 compared to K and D3. Plants on the D2 site showed the highest Pn value, showing a slight difference from plants from the D1 site, which was not confirmed statistically. The significantly lowest Pn value was found in plants from treatment D3, where it was 1.6 $\mu$mol m$^{-2}$ -s$^{-1}$ relative to the control treatment (Table 3). The higher net photosynthetic value in the D1 and D2 treatments may have been influenced by the nutrients (nitrogen, phosphorus, potassium) supplied with the pomace, over and above those supplied as mineral fertilisers. The relationship between increased nutrients and more intensive net photosynthesis was confirmed by Dong et al. [13]. In particular, the additional amount of nitrogen supplied influences higher photosynthetic intensity and lower intercellular $CO_2$ [14].

Transpiration intensity (E) was significantly higher in plants from treatments D1 and D2 compared to K and D3. The higher addition of pomace on the D3 treatment resulted in a significantly lower transpiration intensity compared to the control and both other experimental treatments.

The addition of apple pomace to the soil in treatments D1 and D2 had a beneficial effect on stomatal conductance. A significantly higher stomatal conductance was found in plants compared to sites K and D3. Stomatal conductance measurements taken in plants on treatment D3 showed the significantly lowest stomatal conductance compared to the other experimental treatments, both those with pomace added to the soil (D1 and D2) and the control treatment, indicating the negative effect of too high a level of pomace.

The intercellular $CO_2$ concentration (Ci) was highest in plants on the D2 treatment, and this was a significant difference with respect to the other experimental treatments. Measurements made on the K and D1 sites showed a slightly higher Ci value in plants when pomace was added to the soil, but this was not a statistically confirmed difference. As with the other photosynthesis parameters, the Ci value for plants from treatment D3 was also the lowest.

The value of the water use efficiency (WUE) varied significantly between all sites. The highest value of this parameter was found at site D2. Site D1 had a significantly lower WUE value relative to site D2 but showed a significantly higher WUE value relative to control site K. Treatments D1 and D2 showed that, up to the level of the proportion of apple pomace to soil applied in treatment D2, WUE increased significantly. The lower WUE value of the plants in the D3 facility may have been due to the higher evapotranspiration shown in the experiments conducted. Despite equal irrigation of the plants, water evaporated faster, forcing the plants to use it more sparingly.

**Table 3.** Plant photosynthesis parameters as a function of pomace addition to soil.

| Specification | Pn $(\mu mol\ m^{-2}\ \text{-s})^{-1}$ | E $(mmol\ m^{-2}\ \text{-s})^{-1}$ | gs $(mmol\ H_2O\ m^{-2}\ \text{-s})^{-1}$ | Ci $(\mu mol\ CO_2\ m^{-2}\ \text{-s})^{-1}$ | WUE |
|---|---|---|---|---|---|
| K | 11.4 [b] | 1.6 [a] | 0.202 [b] | 199 [b] | 7.13 [c] |
| D1 | 12.3 [a] | 1.7 [a] | 0.220 [a] | 208 [b] | 7.23 [b] |
| D2 | 12.7 [a] | 1.7 [a] | 0.225 [a] | 225 [a] | 7.47 [a] |
| D3 | 9.8 [c] | 1.4 [c] | 0.187 [c] | 180 [c] | 7.0 [d] |

Different letters (a–d) are significantly different ($\alpha = 0.05$).

The value of the leaf greenness index SPAD for the plants from the experimental treatments studied differed significantly between the K and D3 treatments and the D1 and D2 treatments (Figure 1). The application of apple pomace addition to the soil had a beneficial effect on the SPAD value, with a higher value found in plants from treatment D2 compared to D1, but this was not a statistically significant difference. A higher dose of apple pomace had a lower effect on the SPAD index in plants at site D3 compared to K, but this was not a significant difference. The SPAD index is one tool to determine the nitrogen supply of plants [15]. Apple pomace also contains a certain amount of nitrogen, which can further enrich the available pool of this element in the soil for plants. Apple pomace has a rather diverse composition due to many factors influencing it, such as apple cultivar. According to a study by Krasowska and Kowczyk-Sadowy [16], it can contain between 0.5–1.2% nitrogen in wet weight. The use of equal mineral nitrogen fertilisation in our study and the higher SPAD index may indicate the introduction of an additional dose of nitrogen from pomace used by the plants. The study by Ukalska-Jaruga et al. [17] confirm that an appropriate choice of organic matter source and fertilisation increased the amount of assimilable nitrogen in the soil, thus conditioning the quality of wheat yields. The negative effect of the highest level of pomace applied on the D3 site on the SPAD index may have been due to significant soil acidification and hindered nitrogen uptake by plants due to a less developed root system. According to a study by Krasowska and Kowczyk-Sadowy [16], apple pomace is one of the waste materials which, due to its low pH (in the range of 3.7–4.3), can have a strongly acidifying effect on soil.

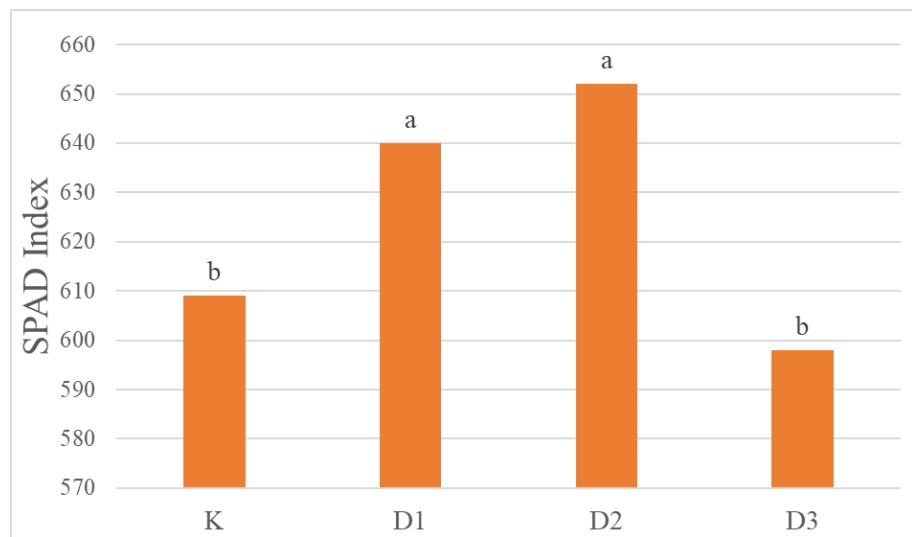

**Figure 1.** SPAD Index average over three years as a function of the level of apple pomace addition to the soil. Different letters (a, b) are significantly different ($\alpha = 0.05$).

### 3.3. Yield Structure Characteristics

The addition of apple pomace to the soil on treatments D1 and D2 resulted in a significant increase in the number of ears compared to the control treatment (K). However, the increased dose of pomace applied on the D3 treatment had a negative effect on the number of ears, and this was a significant difference from both the control treatment and the D1 and D2 treatments (Figure 2). Tillering has a very strong influence on wheat yields because the number of ears per plant depends on it. The tillering process is influenced by many factors, both environmental and the varietal factor itself [18]. The higher number of ears was due to the higher tillering of the plants in sites D1 and D2 (Figure 3). The average tillering in them was 2.3 and 2.4 shoots per plant, respectively, and this was a significant difference from the control treatment, with 2.1 shoots, and the D3 treatment, with 1.4 shoots. A positive effect of the addition of apple pomace to the soil was also found in the weight of grains per ear on the D2 treatment, and this was a statistically significant difference with respect to the other treatments (K, D1 and D3) (Figure 3). Production bushiness, number of ears, and grain weight per ear are the main determinants of yield [19]. The higher number of ears and number of grains per ear also influenced the higher grain weight per spike at sites D1 and D2 compared to the other experimental sites (Figure 4). The lowest grain weight per pot was found for treatment D3 (Figure 5). A similar relationship was found for thousand-grain weight, where D1 and D2 treatments showed significantly higher thousand-grain weights than K and D3 (Figure 6). A difference in thousand-kernel weight was found between treatments D1 and D2, being higher on treatment D2, but this difference was not confirmed statistically. The application of a higher addition of apple pomace to the soil had a negative effect on the weight of one thousand grains, which was confirmed by a statistically significant difference from the control treatment. Ukalska-Jaruga et al. [17] also confirm that the application of an adequate dose of organic matter can influence higher grain weight and yield.

The application of pomace addition on treatments D1 and D2 resulted in significantly higher straw weight in relation to control treatment K. In contrast, the application of pomace addition on treatment D3 had a negative effect on straw weight, and this was a significant decrease in relation to both the control treatment and experimental treatments D1 and D2. The determined harvest index showed no significant differences among the treatments. A significant difference was only found for the higher level of applied pomace on the D3 treatment compared to the D1 and D2 treatments (Table 4).

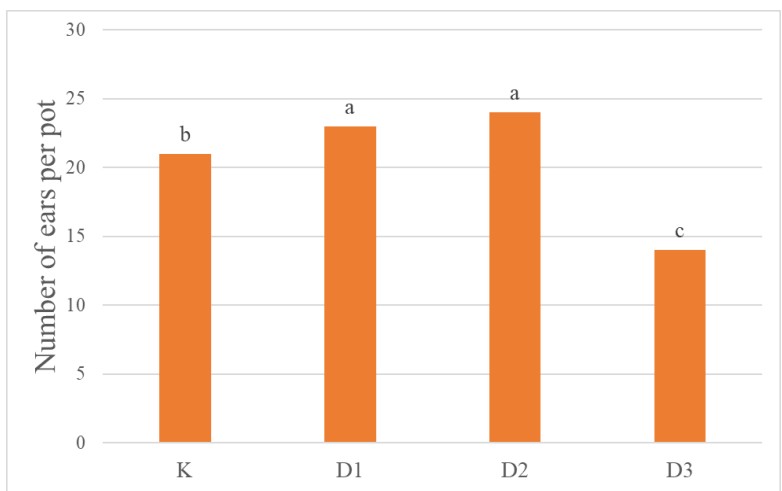

**Figure 2.** Average number of ears per pot depending on the level of apple pomace addition to the soil. Different letters (a–c) are significantly different ($\alpha$ = 0.05).

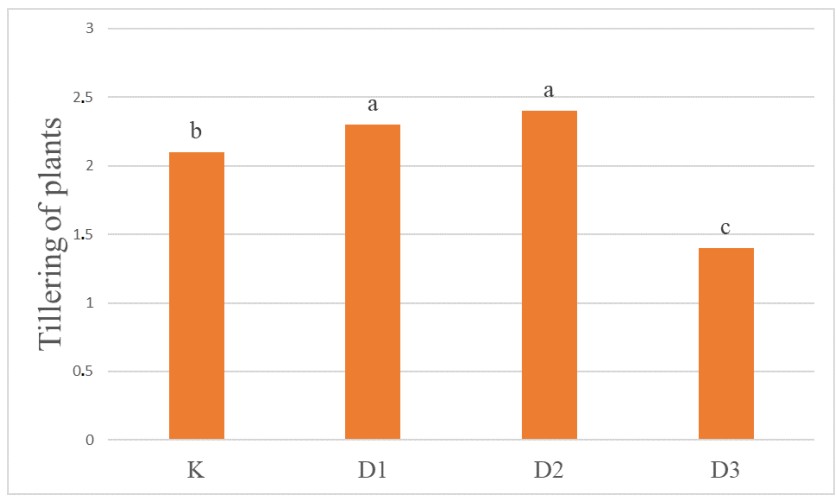

**Figure 3.** Average plant number tillering as a function of the level of apple pomace addition to the soil. Different letters (a–c) are significantly different ($\alpha$ = 0.05).

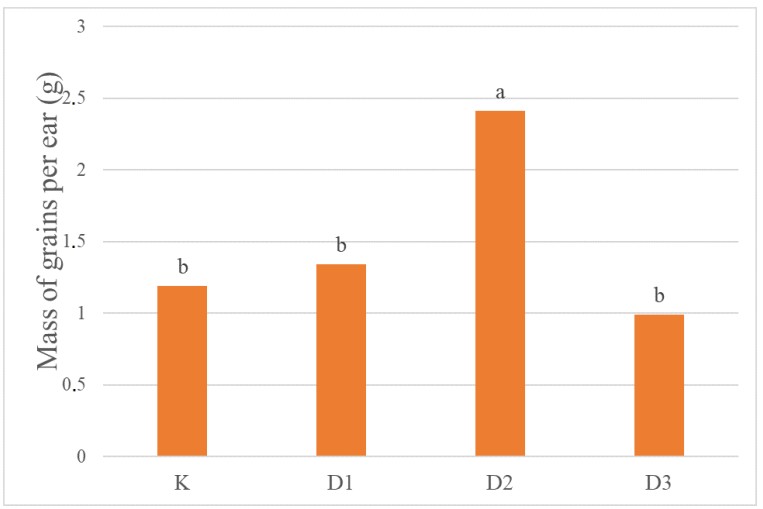

**Figure 4.** Average kernel mass per ear (g) as a function of the level of apple pomace addition to the soil. Different letters (a, b) are significantly different ($\alpha$ = 0.05).

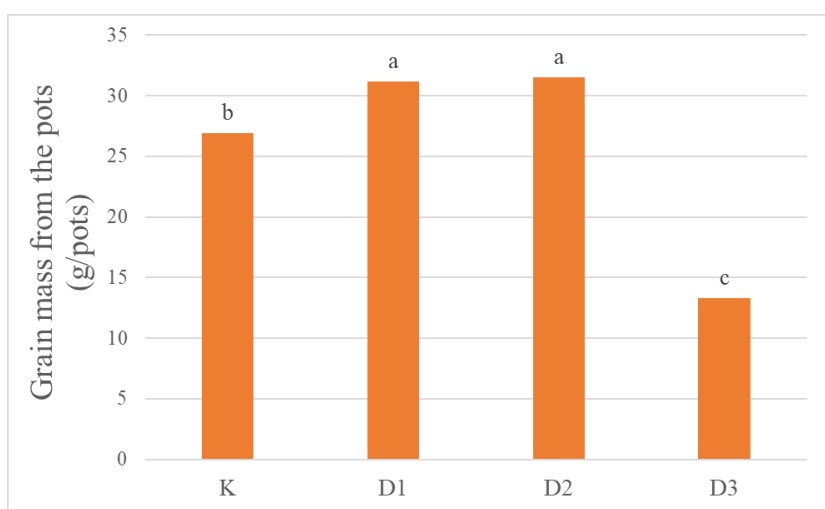

**Figure 5.** Average mass of grains per pot (g) as a function of the level of addition of apple pomace to the soil. Different letters (a–c) are significantly different ($\alpha$ = 0.05).

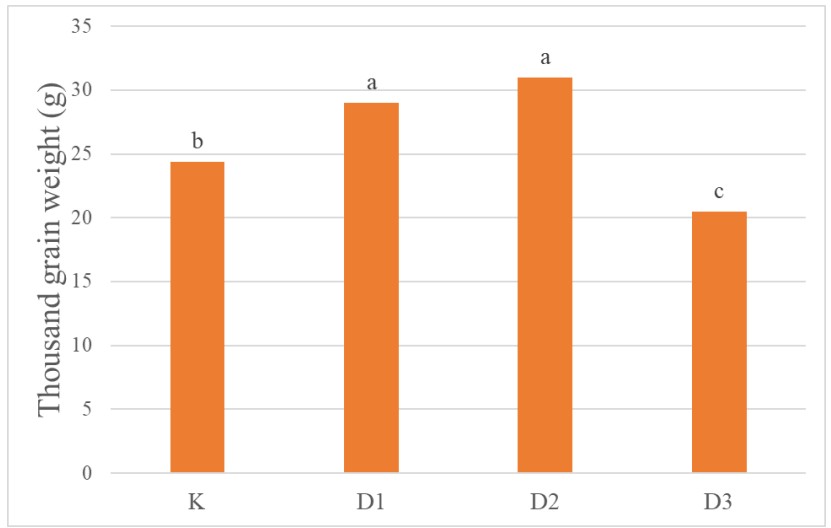

**Figure 6.** Thousand kernel weight as a function of the level of addition of apple pomace to the soil. Different letters (a–c) are significantly different ($\alpha$ = 0.05).

**Table 4.** Grain weight, straw and harvest index as a function of the level of apple pomace addition to the soil.

| Specification | Grain Weight (g) | Weight of Straw (g) | Harvest Index (%) |
|:---:|:---:|:---:|:---:|
| K | 26.9 [b] | 32.19 [b] | 33 [ab] |
| D1 | 31.2 [a] | 33.44 [a] | 34 [a] |
| D2 | 31.5 [a] | 34.68 [a] | 36 [a] |
| D3 | 13.3 [c] | 28.6 [c] | 30 [b] |

Different letters (a–c) are significantly different ($\alpha$ = 0.05).

## 4. Conclusions

Apple pomace, as a waste product added to the soil, can provide a source of organic matter and have a positive effect on soil properties and the yield of spring wheat. However, it has been shown that the optimum pomace addition rate, in a pot experiment, for a beneficial effect on plants and soil is the rate applied on the experimental site D2 under field conditions of 2 t·ha$^{-1}$. This proportion of apple pomace has a positive effect on soil

properties and on plant physiological indicators, structural characteristics, and final grain yield. As the study was conducted under model conditions, further research under micro plot or field conditions is necessary. The possibility of using pomace and its optimal doses should also be checked for other types of crops.

**Author Contributions:** Conceptualization, M.R. and M.W.; methodology, M.R. and J.G.; software, M.R.; validation, M.R., M.W. and J.G.; formal analysis, M.R.; investigation, M.R.; resources, M.R.; data curation, M.R.; writing—original draft preparation, M.R. and M.W. writing—review and editing, M.R.; visualization, M.R.; supervision, J.G.; project administration, M.R. All authors have read and agreed to the published version of the manuscript.

**Funding:** This research received no external funding.

**Data Availability Statement:** The data presented in this study are available on request from the first author.

**Acknowledgments:** We would like to thank Mariola Staniak and Anna Stępień-Warda for providing a device for measuring photosynthesis and the possibility of its implementation. We would like to thank Jolanta Bojarszczuk for taking measurements of soil respiration.

**Conflicts of Interest:** The authors declare no conflict of interest.

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
