# Peer review of "Effect Application of Apple Pomace on Yield of Spring Wheat in Potting Experiment"

_agronomy, doi:10.3390/agronomy13061526_

Round 1
Reviewer 1 Report
Dear editor,
the study focused on the effects of soil application of apple pomace on yield of spring wheat. The study was well designed and suitable for publishing in the Agronomy journal but the authors should modify some corrections. Therefore, my final recommendation is "Major revision".
Abstract
- Please add one or two sentence in the first section of abstract as background of the study.
- Underscore the scientific value-added to your paper in your abstract. Your abstract should clearly state the essence of the problem you are addressing, what you did and what you found and recommend. That will help a prospective reader of the abstract to decide if they wish to read the entire article.
- The values should be added in abstract section.
- What is the final recommendation for wheat growers based on the experiment results?
Introduction
This section should be revised.
- The novelty should be highlighted.
- The hypothesis should be added at the end of this section.
Materials and methods
- Some methods of measured traits such as WUE and etc. were not added in this section.
Results
- The unit of measured traits should be added in tables.
- The increasing or decreasing percentages between treatments should be added in results section.
- I recommended to authors for adding the macro and micro-nutrient concentration of wheat in different treatments.
- Please add the correlation among main measured traits of wheat.
Discussion
I can’t see any discussion in this manuscript. Please add the main reason for increasing or decreasing the measured traits.
Conclusion
- This section is repetitive and should be rewritten.
- Please make sure your conclusions' section underscores the scientific value-added of your paper, and/or the applicability of your findings/results. Highlight the novelty of your study.
- What suggestions do you have for future research in this field?
Minor editing of English language required
Author Response
Dear Reviewer, thank you very much for taking the time to read the work and very valuable comments that will help improve the final version of the article. According
As suggested by you, the summary has been revised and a final recommendation on the use of pomace by wheat growers has been added.
We have corrected the introduction and added a research hypothesis at the end.
The WUE measurement is indicated in the Methodology as calculated from the formula given in lines 112-113
The discussion was conducted along with the results. because there is little available literature on the application of pomace to the base, we relied on the small amount of literature that concerns the results presented by us.
Conclusions from our experience as a model, in defined and controlled conditions, we consider sufficient. We cannot draw any more conclusions. However, we have already included recommendations as to the necessity of repeating the tests in field conditions.
Reviewer 2 Report
The research idea is very interesting, while there are a lot of serious methodological flaws in the article.
1. the title should be changed. You can not write that it was applied to the soil (it is not a field). There should be a mention that it is a potting experiment.
2. at the beginning of the introduction should quote the literature on the volume of apple production in Poland.
3. hypothesis and objective need a thorough rewrite
4. chemical composition should be given in g/kg-1
5. on what substrate (soil) the study was established. chemical composition
6. how the pots were watered
7. 30 individual measurements are needed to measure SPAD. Flag leaves were 10. This means that several measurements were made on one leaf?
8. why the statistical analysis did not take into account the effect of years of study?
9. from what years of the study are the results? Why was the effect of years on the studied traits not given. The methodology states that there were 3 years of research. In the tables it is not clear what year of research or the average of years?
10. figure 3 - no unit
11. figure 4 - weight or mass?
12. figure 5 - weight or mass?
13. harvest index of yield?
14. why mineral fertilization rates are not given in kg/ha?
15. apple pomace - what variety. Does variety affect the chemical composition of apple pomace.
At present, the article presented is not suitable for publication. It requires extensive revision and rewriting and resubmission to the publisher.
My recomendation is to reject it in its present form.
Author Response
Thank you very much for taking the time to review the work and for valuable comments that will allow for its improvement.
- As you suggested, it has been added in the title that this is a mdoel experiment conducted as a potted one.
- Unfortunately, the scale of apple production is only an estimate based on data available from producers and industrial processors of apples and is about 3.9 million, but these are variable data depending on the apple yield itself related to weather conditions and possible spring frosts. We do not have official data from the literature and cannot cite them.
- The hypothesis and the aim of the work have been changed and rewritten
- The values have been changed according to your suggestion from % to g/kg-1
- The top layer of soil (up to 20cm) from a wheat field was used in the experiment the pseudopodsolic soil typical of the region with extractable phosphorus (P: 9.54 mg kg−1), exchangeable potassium (K: 12.0 mg kg−1), and pH KCl 6.4.
- Watering was carried out automatically using the set system, twice a day 120 ml of water per vases (total 240 ml per day)
- In the experiment, we had 40 plants in total on one experimental object, 4 pots of 10 plants each, which gave a total of 40 plants and forty flag leaves.
- The effects of the years were to check the influence of the same pomace from different years. Because it was an experiment in model conditions, the same irrigation, soil and wheat variety, it turned out that there is no interaction of years and the presented results are a synthesis of three years of research.
- All test results in studies with the same trend, therefore the statistical analysis was carried out together for the experiment from three years of research and the results are their average.
- it has been clarified that it is the number of average tiller
- and 12 it has been clarified that it is the weight of the grain
- in scientific literature and other scientific publications there is only a harvest index
- Mineral fertilization was given in such a unit because it is not a field experiment but a pot experiment. Therefore, we did not calculate the amount of micronutrients per field area, but only the weight of pomace that is given to the field, so that wheat growers could use it on the field if they wanted
- The pomace used was from an apple of the ligol variety
Round 2
Reviewer 2 Report
Thank you very much for responding. I disagree with three of the answers.
In the experiment, we had 40 plants in total on one experimental object, 4 pots of 10 plants each, which gave a total of 40 plants and forty flag leaves.
1. I know very well that there were 10 plants in each pot, four pots are 40 plants. This is true. So out of these 40 plants we have only one SPAD measurement. Where did you get the values for statistical analysis from when we only have one measurement. This is a methodological error of the experiment.
The effects of the years were to check the influence of the same pomace from different years. Because it was an experiment in model conditions, the same irrigation, soil and wheat variety, it turned out that there is no interaction of years and the presented results are a synthesis of three years of research.
2. The answer is unacceptable. Skora authors assume that there is no impact of years then why do they conduct a study in three years. It is a waste of time to do such research. So the results can be accelerated.
In scientific literature and other scientific publications there is only a harvest index
Yes I agree with this answer, while we still have an indicator of, for example, nitrogen harvesting.
He suggests turning years, into series. Then the stayist analysis must be corrected. After a thorough rewrite, the article must be sent for a second review
Author Response
Thank you very much for more valuable comments. Answering your questions and concerns below:
- In fact, and we had 9 SPAD results in total, the results obtained were treated as a series, as you recommended in your last comment. This is how the statistical analysis was obtained.
- We assumed no influence of years as a weather factor. Since this was an experiment under strictly controlled conditions of irrigation, soil, fertilization, strains, all these factors were the same. Only the variability concerned apple pomace that came from different years, from each of the examined years, i.e. 2019, 2020 and 2021. We checked whether there was a different impact from different years, because pomace is made of raw material which is apple, which may have a different chemical composition depending on the weather. Hence the long-term experiment to check the repeatability of the results. The fact that the results are the same in the same tendency of pomace influence we found only after 3 years of the experiment. After one year, this value was insufficient and the effect itself could be undermined.